# Functional large-conductance calcium and voltage-gated potassium channels in extracellular vesicles act as gatekeepers of structural and functional integrity

Shridhar Sanghvi[1,2,10], Divya Sridharan[3,10], Parker Evans[4], Julie Dougherty[3], Kalina Szteyn[2], Denis Gabrilovich[1], Mayukha Dyta[2], Jessica Weist [3], Sandrine V. Pierre[5], Shubha Gururaja Rao[6], Dan R. Halm[7], Tingting Chen[8], Panagiotis S. Athanasopoulos[8], Amalia M. Dolga [8], Lianbo Yu[9], Mahmood Khan[3] ✉ & Harpreet Singh [1,2] ✉

Extracellular vesicles (EVs) are associated with intercellular communications, immune responses, viral pathogenicity, cardiovascular diseases, neurological disorders, and cancer progression. EVs deliver proteins, metabolites, and nucleic acids into recipient cells to effectively alter their physiological and biological response. During their transportation from the donor to the recipient cell EVs face differential ionic concentrations, which can be detrimental to their integrity and impact their cargo content. EVs are known to possess ion channels and transporters in their membrane but neither the function nor the role of these channels in EVs is known. In this study, we discover a functional calcium-activated large-conductance potassium channel ($BK_{Ca}$) in the membrane of EVs. Furthermore, we establish that $BK_{Ca}$ is essential for the structural and functional integrity of EVs. Together, these findings establish the critical role of ion channels such as $BK_{Ca}$ in functioning as gatekeepers and maintaining EV-mediated signaling.

Extracellular vesicles (EVs) are nanomembrane vesicles that are 40–1000 nm in size, surrounded by a lipid bilayer, and secreted by most cell types. They are implicated in the delivery of bioactive cargo such as protein, nucleic acids, lipids, metabolites, and, more recently drugs[1]. Their recent advent in therapeutic and diagnostic applications for the treatment of various diseases, including cardiovascular

dysfunction, cancer, and neurological disorders, made them highly promising tools in medicine. EVs are endosomal in origin and contain 9769 proteins, 3408 mRNA, 2838 miRNA, and 1116 lipids[2]. The efficient delivery and exchange of these heterogeneous components is essential for intercellular communication and the designing of EV-based therapeutics. Since the EVs are released as multivesicular bodies (MVB)

[1]Department of Molecular Cellular and Developmental Biology, The Ohio State University, Columbus, OH, USA. [2]Department of Physiology and Cell Biology, The Ohio State University Wexner Medical Center, Columbus, OH, USA. [3]Division of Basic and Translation Research, Department of Emergency Medicine, The Ohio State University Wexner Medical Center, Columbus, OH, USA. [4]Department of Mechanical and Aerospace Engineering, The Ohio State University, Columbus, OH, USA. [5]Department of Biomedical Sciences, Joan C. Edwards School of Medicine, Marshall University, Huntington, WV, USA. [6]Department of Pharmaceutical and Biomedical Sciences, The Raabe College of Pharmacy, Ohio Northern University, Ada, OH, USA. [7]Department of Neuroscience, Cell Biology, and Physiology, Wright State University, Dayton, OH, USA. [8]Department of Molecular Pharmacology, Groningen Research Institute of Pharmacy, University of Groningen, Groningen, NE, USA. [9]Department of Biomedical Informatics, The Ohio State University, Columbus, OH, USA. [10]These authors contributed equally: Shridhar Sanghvi, Divya Sridharan. ✉e-mail: Mahmood.khan@osumc.edu; Harpreet.singh@osumc.edu

by exocytosis into the extracellular environment, the presence of ion channels is critical for their survival and function. However, the precise mechanism of how EVs handle the differential ionic environment, as well as the physiological role and identity of their ion channels, is not established.

To elucidate the role of ion concentrations in EVs integrity, we examined the Nernst potential for major ions. The Nernst potential between a typical cellular and extracellular environment is 134 mV for calcium ions, 65 mV for sodium ions, −92 mV for potassium, and −45.5 mV for chloride ions[3]. We hypothesized that if EVs can handle the huge ionic potential difference, then they would need a rapid ion transport mechanism. To regulate ionic homeostasis, 72 unique ion channels and 107 transporters or exchangers are present in EVs in the ExoCarta database (Supplementary Table 1). Due to the size of the EVs, establishing the functional identity and biophysical properties of ion channels has been challenging. In this study, we sought to identify the functional ion channel involved in maintaining ionic homeostasis that governs the size and number of EVs.

In physiology, the univalent cation with the highest gradient across the plasma membrane are $K^+$ ions. Intracellular $K^+$ concentration is ~140 mM, whereas the extracellular concentration is ~4 mM. To counter the osmotic stress of the intracellular-to-extracellular $K^+$ gradient, we reasoned EVs to possess a rapid $K^+$ transportation mechanism. One of the large-conductance ion channels is a calcium-activated and voltage-gated potassium channel ($BK_{Ca}$, encoded by a gene, $Kcnma1$), which conducts $K^+$ ions across the cell membrane[4]. $BK_{Ca}$ channel activity and altered expression have been implicated in the regulation of cell volume in an altered osmotic environment[5,6]. Therefore, we characterized the functional properties of $BK_{Ca}$ in EVs and established their role in handling differential potassium concentrations. Moreover, different cellular stresses alter the molecular cargo of EVs, affecting intracellular signaling and mediating several biological and cellular processes[1]. $BK_{Ca}$ channel is shown to regulate the cargo in EVs during an inflammatory response[7]. We further investigated the miRNA profile in EVs isolated from wild-type ($Kcnma1^{+/+}$), and null mutant ($Kcnma1^{-/-}$) mice. Our results indicate the presence of functional $BK_{Ca}$ channels in EVs, which are crucial for maintaining their structural integrity and survival in differential [$K^+$] ionic gradients.

## Results

### Presence of iberiotoxin-sensitive potassium currents in EVs

A growing body of evidence indicates ion channel proteins are present in EVs and they can be transported by these nanovesicles[8–11]. To determine whether functional channels are present in EVs, we used a combination of imaging and electrophysiology to identify ion channels in EVs. We isolated EVs from the plasma of C57BL/6NCrL mice and investigated their purity and integrity (Fig. 1a, i). The key challenges to determining the presence of functional channels in EVs are their small size and current experimental limitations, which make it difficult to incorporate canonical approaches such as patch clamp. Hence, we addressed this limitation by applying a novel measurement system called Near Field Electrophysiology (NFE) that relies on chronoamperometric detection of extracellular $K^+$ concentration in the diffusion-limited region around the EVs (Fig. 1b and Supplementary Fig. 1A-C)[12–15]. A cell's *electrical state* reflects a collection of chemoelectrical variables. This electrical state captures the evolution of ionic currents and concentration due to changes in extracellular concentration. Our NFE approach indicates a presence of $K^+$ channels in intact EVs, and 45% of them are sensitive to Iberiotoxin (IBTX) (Fig. 1c). Since (IBTX) specifically blocks $BK_{Ca}$ channels, we estimated 2 functional channels (assuming typical single-channel conductance of 300 pS with 50% open probability) in a single EV.

Though NFE provides direct evidence of the existence of IBTX-sensitive $K^+$ channels in intact EVs, it is difficult to measure the single-channel activity of ion channels with this approach. Therefore, we used

a traditional planar bilayer technique where purified EV membranes were reconstituted in artificial lipid bilayers. In $KCH_3SO_3$ solutions (500 mM: 50 mM:: cis: trans), the single-channel conductance of $335 \pm 15$ pS (n = 8 of 13 successful reconstitutions) was recorded (Fig. 1d, f). Amplitude histograms show an open probability of 50% at +40 mV (Fig. 1e). The addition of IBTX (100 nM) decreased open probability (Po) by 75% (Fig. 1g, h). Membranes reconstituted from EVs obtained from $Kcnma1^{-/-}$ mice did show a $K^+$ current, albeit with a single channel conductance of $110 \pm 10$ pS (Supplementary Fig. 2A, B). These results indicate the possible presence of other $K^+$ channels in EVs, in addition to $BK_{Ca}$ channels.

### Potassium ions maintain EV integrity

EVs originate from intracellular membranes and are predicted to have similar ionic concentrations as cytosol. As they enter the extracellular environment, they face a large ionic variability. These vesicles need a rapid ion exchange mechanism to adapt to the extracellular ion concentrations. Potassium is the most abundant univalent cation in the cytosol[16] and its concentration falls ~30-fold as EVs are secreted into the extracellular environment. To determine the contribution of $K^+$ ions to EV size and integrity, we isolated EVs and performed nanoparticle-tracking analysis (NTA) to determine their numbers and size distribution (Fig. 1j–o).

We measured the mean size of EVs in intracellular $K^+$ (145 mM) concentrations as well as extracellular $K^+$ (4 mM) concentrations to establish the impact of variable $K^+$ concentrations on EVs. As compared to PBS, the mean size of EVs decreased from $177.43 \pm 2.27$ nm (control, PBS) to $161.36 \pm 9.33$ nm in extracellular 4 mM $K^+$, but EVs size increased to $217.06 \pm 12.59$ nm in intracellular 145 mM $K^+$ (Fig. 1j). We also measured EV size in NS1619 and IBTX. As compared to PBS (Fig. 1j), where the size was $161.36 \pm 9.33$ nm, EVs treated with NS1619 (activator) were $167.03 \pm 2.43$ nm (Fig. 1k) in size, whereas treatment IBTX (blocker) decreased their size to $152.4 \pm 4.82$ nm (Fig. 1k). EVs isolated from $Kcnma1^{-/-}$ mice showed no difference in size between intracellular and extracellular $K^+$ concentrations (Fig. 1l). We further compared the mean size of EVs isolated from $Kcnma1^{+/+}$ and $Kcnma1^{-/-}$ mice (Fig. 1m). The size of EVs isolated from $Kcnma1^{-/-}$ mice was smaller than EVs isolated from $Kcnma1^{+/+}$ mice (Fig. 1m). However, when we quantified the total number of EVs (Fig. 1n), we found that $Kcnma1^{-/-}$ mice had almost double the number of EVs compared to $Kcnma1^{+/+}$ mice (Fig. 1o). These results indicate the size dependence of EVs on $K^+$ ions in the extracellular environment.

### Presence of $BK_{Ca}$ channels in EVs

To visualize the presence of a $BK_{Ca}$ channel in EVs, we isolated EVs from the plasma of $Kcnma^{+/+}$ mice and assessed the presence of $BK_{Ca}$ protein via western blotting (Fig. 2a). EVs probed with a highly specific anti-$BK_{Ca}$ antibody (Supplementary Fig. 3e) showed a presence of $BK_{Ca}$ in mice at ~115 kDa (Fig. 2a). EVs were labeled with PKH67 (Fig. 2b, f) and probed with a highly specific anti-$BK_{Ca}$ antibody (Fig. 2c vs. g). Positive signals coinciding with PKH67 labeled EVs were observed in $Kcnma^{+/+}$ mice ($72 \pm 5\%$, Fig. 2b–e), but those from $Kcnma1^{-/-}$ mice, had no $BK_{Ca}$ specific signal (Fig. 2f–i). We further purified EVs from the culture media of mesenchymal stem cells (MSCs). The isolated EVs labeled with PKH67 (Supplementary Fig. 2A) were spread on poly-l-lysine-coated coverslips and probed with a highly specific anti-$BK_{Ca}$ antibody (Supplementary Fig. 3D). All the EVs showed labeling for $BK_{Ca}$ channels (Supplementary Fig. 3A–D). The localization percentage was $95 \pm 2\%$ (n = 5 preparations).

As EVs are known to transport proteins[17], we also tested whether EVs isolated from the plasma of $Kcnma^{+/+}$ and $Kcnma1^{-/-}$ mice can be taken up by cells. EVs were loaded with PKH67 and incubated for 8 hr with human-induced pluripotent stem cells derived cardiomyocytes (hiPSC-CMs, Fig. 2j–u). Cells were labeled with WGA, fixed, permeabilized, and probed with an anti-$BK_{Ca}$ antibody. The colocalization

between the BK$_{Ca}$ channel and EVs was approximately $65 \pm 7\%$ (Fig. 2j–n), which was similar to EVs isolated from the wild-type (Fig. 2b–e). EVs isolated from *Kcnma1$^{-/-}$* mice showed signal in only $10 \pm 2\%$ of the total PKH67 population (Fig. 2p–u), which could originate from native BK$_{Ca}$ being loaded in vesicles while inside cells or non-specific signal from secondary antibodies. The presence or absence of BK$_{Ca}$ does not affect the uptake of EVs into cells. Next, we probed the possible regulation of K$^+$ homeostasis in EVs. Since the presence of Sodium Potassium ATPase (Na-K-ATPase) in EVs has been reported in mass spectrometry analysis of EVs[18,19], we also tested the presence of Na-K-ATPase in plasma-derived EVs. Our data showed the presence of BK$_{Ca}$ along with Na-K-ATPase in freshly isolated human plasma-derived EVs (Supplementary Fig. 4A). ATP is vital for the activation of Na-K-

ATPase, and we did detect a significant amount of ATP in freshly isolated EVs from plasma (Supplementary Fig. 4B).

## BK$_{Ca}$ channels regulate the function and contents of the EVs

To understand whether the presence or absence of BK$_{Ca}$ on EVs influences the content of EVs, we performed nano-string miRNA analysis on the cargo in plasma EVs isolated from *Kcnma1$^{+/+}$*, and *Kcnma1$^{-/-}$* mice. In total, 600 miRs were analyzed, and 42 hits were obtained (Fig. 3a). miRNAs identified were distributed into two groups (1) 28 miRNAs, downregulated (Fig. 3a), and (2) 14 miRNAs upregulated in EVs from *Kcnma1$^{-/-}$* vs. *Kcnma1$^{+/+}$* mice (Fig. 3a). Among miRNAs known to regulate cardioprotection, in the absence of BK$_{Ca}$, miR-19a, miR-23a, and miR-486 were downregulated (Fig. 3b–d), and miR-let-7g, miR-15a,

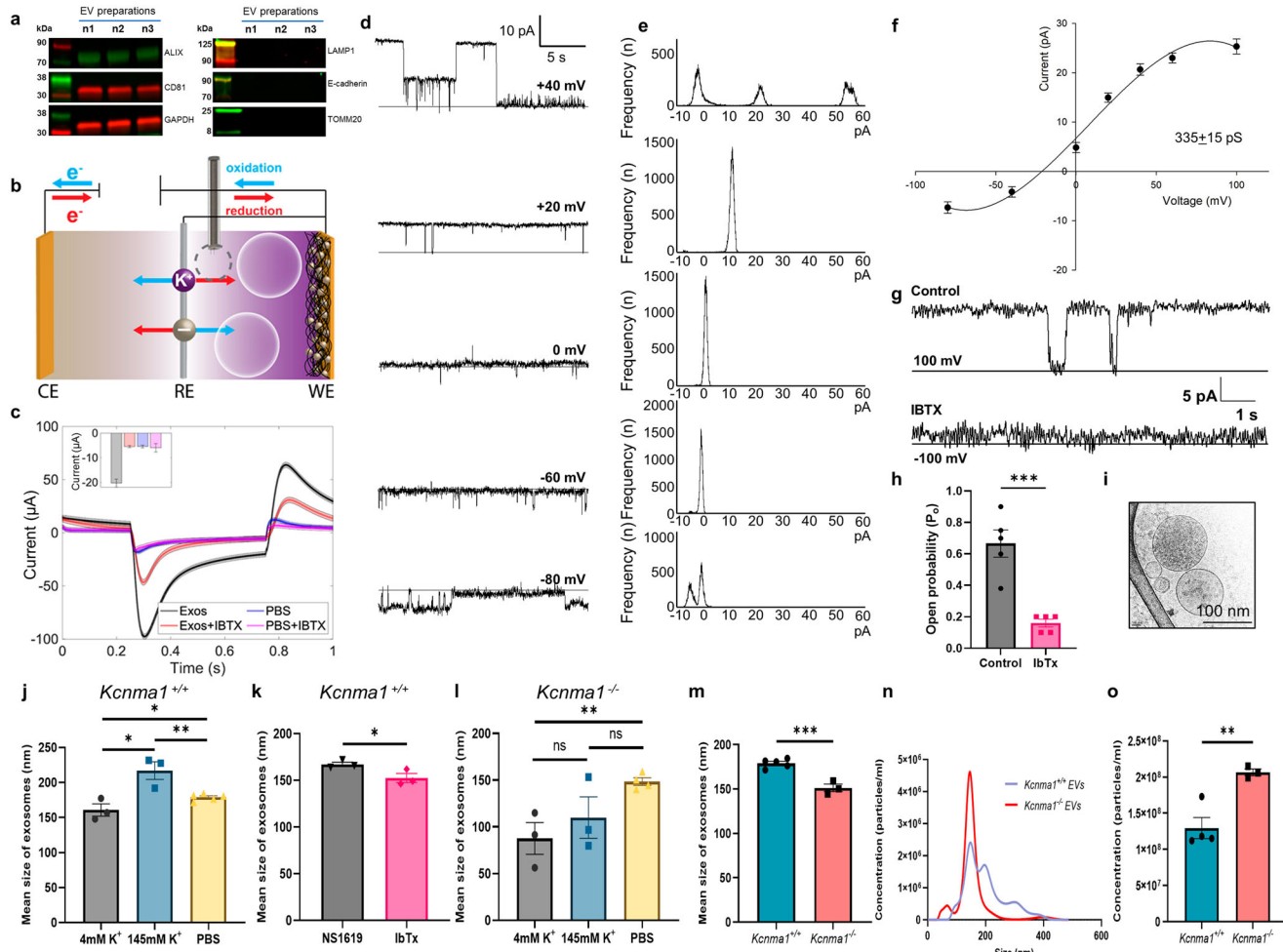

**Fig. 1 | Presence of iberiotoxin-sensitive K$^+$ currents in extracellular vesicle membranes. a** Characterization of plasma-derived EVs isolated from C57BL/6NCrL mice using Western blots, which were positive for EV markers: ALIX, CD81, and GAPDH. The EV isolation was devoid of contamination from the lysosome marker (LAMP1), plasma membrane marker (E-cadherin), and mitochondrial marker (TOMM20). **b** Scheme showing the experimental approach of Near field electrophysiology (NFE) with/without EVs. **c** Traces indicate K$^+$ current increases in the presence of EVs (black) and were inhibited in the presence of iberiotoxin (orange). **d** Single channel currents of K$^+$ channel were recorded from lysed EV membranes in 500:50 mM *cis* versus *trans* asymmetrical KCH$_3$SO$_3$ containing 1 mM DTT. Representative traces of K$^+$ currents at positive and negative holding potentials. The solid line indicates the closed state of the channel. **e** The frequency histogram of corresponding traces from **d**. **f** Current versus voltage (I/V) curve of potassium channel in asymmetrical KCH$_3$SO$_3$. The single-channel conductance was $335 \pm 15$ pS (n = 8). Error bars are mean $\pm$ SD. **g** Representative traces of K$^+$ current blocked by IBTX.

**h** Percentage block of K$^+$ current in the presence or absence of IBTX (n = 5). **i** Cryo-TEM images of the purified plasma-derived EVs isolated from C57BL/6NCrL. **j** EVs isolated from *Kcnma1$^{+/+}$* mice were incubated in physiological [K$^+$] extracellular [4 mM], and intracellular [145 mM] concentrations and compared to PBS. Increased [K$^+$] is directly proportional to the size of EVs. **k** *Kcnma1$^{+/+}$* ionophore NS1619 increases the size of EVs compared to iberiotoxin. **l** EVs isolated from *Kcnma1$^{-/-}$* incubated in extracellular [4 mM], and intracellular [145 mM] concentrations compared to PBS. *Kcnma1$^{-/-}$* EVs lose K$^+$ handling capacity. **m** Mean size of EVs isolated from *Kcnma1$^{+/+}$* mice compared to *Kcnma1$^{-/-}$* mice. **n, o** The concentration of circulating plasma EVs in *Kcnma1$^{+/+}$* and *Kcnma1$^{-/-}$* mice. **h** Data represented as mean $\pm$ SEM, and significance was determined by a two-tailed Student's *t*-test from independent biological replicates; *$p \leq 0.05$. **j–o** Statistical analyses were carried out using a two-tailed Student's *t*-test (n = 3-5 biological repeats). Data are presented as mean values $\pm$ SEM with the individual biological samples shown: *$p \leq 0.05$, **$p \leq 0.01$, ***$p \leq 0.001$.

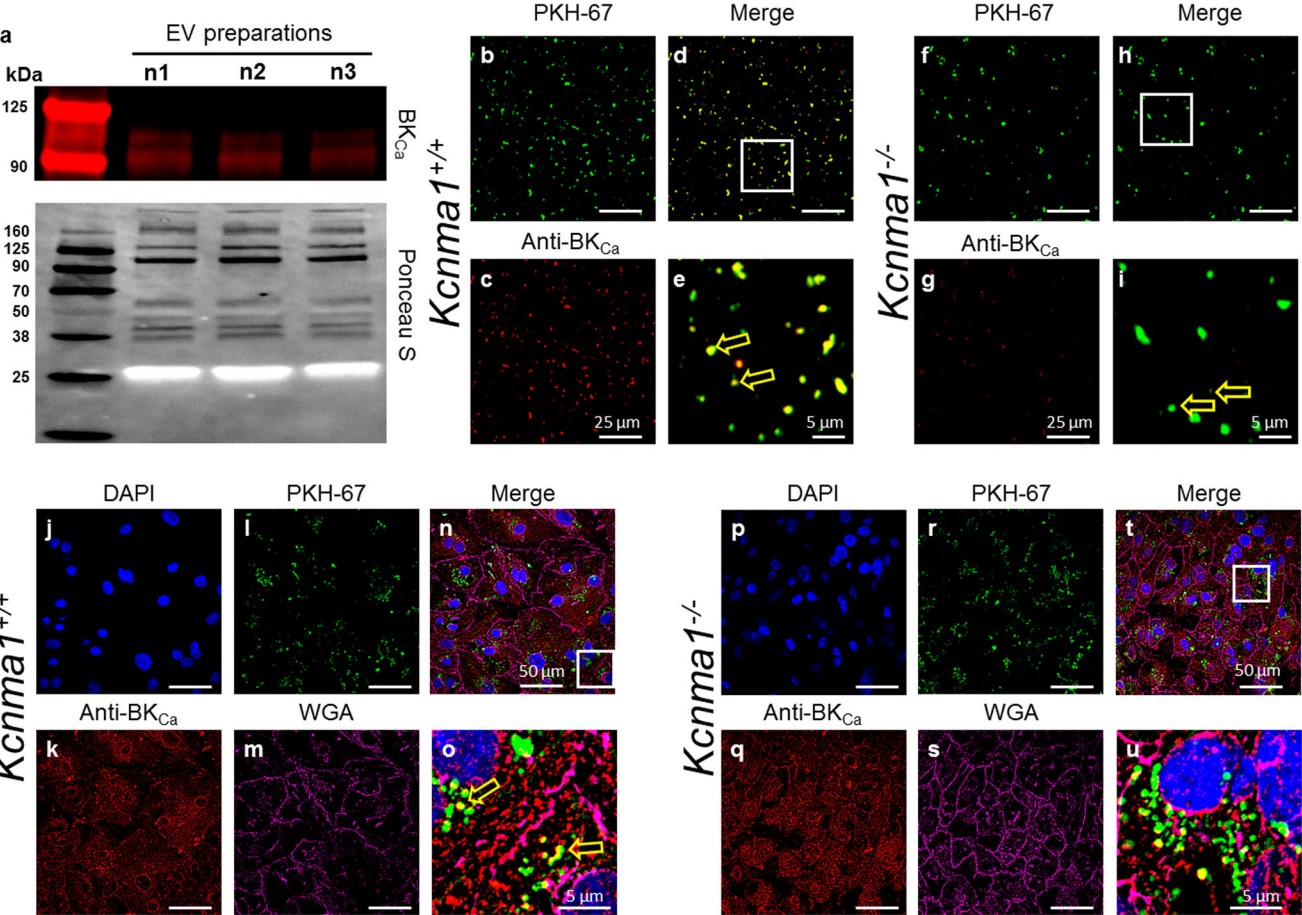

**Fig. 2 | Presence of BK$_{Ca}$ in EVs. a** Representative Western blot showing the presence of BK$_{Ca}$ and protein loading control (Ponceau S) in EVs isolated from plasma of mice (n = 3). Purified EVs from *kcnma1$^{+/+}$* mice (**b–e**) (shown as green, **d**) and *kcnma1$^{-/-}$* mice (**f–i**) labeled with pkh67 (green) and anti-BK$_{Ca}$ (red). Human iPSC-derived cardiomyocytes (hiPSC-CM, **j–u**) were incubated with EVs isolated from *kcnma1$^{+/+}$* and *kcnma1$^{-/-}$* labeled pkh67 (green, **l**, and **r**, respectively). hiPSC-CM

were loaded WGA (magenta, **m**, and **s**) and fixed for immunocytochemical analysis. hiPSC-CM were permeabilized and labeled with anti-BK$_{ca}$ antibodies (red, **k**, and **q**) and DAPI (blue, **j**, and **p**). All images were merged in **n** (**j–m**) and **t** (**p–s**). **o** and **u** are zoomed regions from **n** and **t**, respectively. bk$_{ca}$ was present in evs isolated from *kcnma1$^{+/+}$* (yellow arrow) but not in *kcnma1$^{-/-}$* mice.

and miR-144 were upregulated (Fig. 3e–g). Our nano string miRNA analysis implicates a possible involvement of BK$_{Ca}$ in determining the cargo content of EVs.

### BK$_{Ca}$ channels in EVs attenuate oxidative stress-mediated effects on hiPSC-CMs

Since the presence of BK$_{Ca}$ positively correlates to the enrichment of cardioprotective miRNAs like miR-23a, miR-19a, let-7g, and miR-144 (Fig. 3a–g), we evaluated the efficacy of EVs isolated from *Kcnma1$^{+/+}$* mice in mediating cardioprotection against oxidative stress in hiPSC-CM. HiPSC-CMs were incubated with a serum-free and glucose-free medium for 16 hr with or without EVs (5000 EVs/cell) isolated from either *Kcnma1$^{+/+}$* or *Kcnma1$^{-/-}$* mice plasma. As shown in Fig. 3h, at 10 and 20 min after the treatment with H$_2$O$_2$, the hiPSC-CMs without EVs or EVs from *Kcnma1$^{-/-}$* mice showed a decreased beat rate by 81.70 ± 16.70% as compared to hiPSC-CMs with EVs from *Kcnma1$^{+/+}$* mice which decreased by 20.87 ± 15.28%. These results indicate a reduction in viable hiPSC-CMs on treatment with H$_2$O$_2$, which can be rescued by EVs from *Kcnma1$^{+/+}$* mice. hiPSC-CMs treated with EVs *Kcnma1$^{+/+}$* mice showed viable and functional cells even after 20 min of H$_2$O$_2$ treatment.

As we observed changes in miRNA packaging in EVs (Fig. 3a) between *Kcnma1$^{+/+}$* and *Kcnma1$^{-/-}$* and the impact of EVs on hiPSC-CM function, we next tested whether these miRNAs directly impact

cardioprotection in hiPSC-CMs. We performed the H$_2$O$_2$ assays, as shown in Fig. 3h, after pre-incubating the cells with mimics or inhibitors for miR-15a and miR-23a. We observed that miR-15a is enriched in EVs isolated from the plasma of *Kcnma1$^{-/-}$* mice as compared to *Kcnma1$^{+/+}$* mice, and miR-15a mimic had a similar deleterious impact on hiPSC-CMs as observed for EVs lacking BK$_{Ca}$ channels (Fig. 3j, k). The beat rate of hiPSC-CM treated with miR-15a mimic reduced by 77.83 ± 22.16% whereas miR-15a inhibitor-treated CMs reduced by 42 ± 33.56% within 20 min of H$_2$O$_2$ treatment. In contrast, the miR-15a inhibitor had a protective effect against H$_2$O$_2$ treatment (Fig. 3j, k). We also tested the impact of miR-23a mimic and miR-inhibitor on hiPSC-CMs (Fig. 3j, k). The beat rate of hiPSC-CM treated with miR-23a mimic reduced by 20.82 ± 13.74% and miR-23a inhibitor by 70.67 ± 29.32% within 20 min of H$_2$O$_2$ treatment. On electrophysiology analysis, the miRNA 23a mimic preserved the beat rate of hiPSC-CMs, but the miRNA 23a inhibitor had a deleterious impact (Fig. 3j, k). Consistent with the change in hiPSC-CM function, we observed a significant increase in the LDH release following H$_2$O$_2$ treatment in hiPSC-CMs pre-treated with miRNA-15a mimic and miRNA-23a inhibitor. On the other hand, hiPSC-CMs pre-treated with miRNA-15a inhibitor and miRNA-23a mimic did not show an increase in LDH release, indicating the involvement of these miRNAs in cardioprotection (Fig. 3l). These findings indicate that EVs containing BK$_{Ca}$ channels play an active role in protecting cardiac cells from oxidative stress.

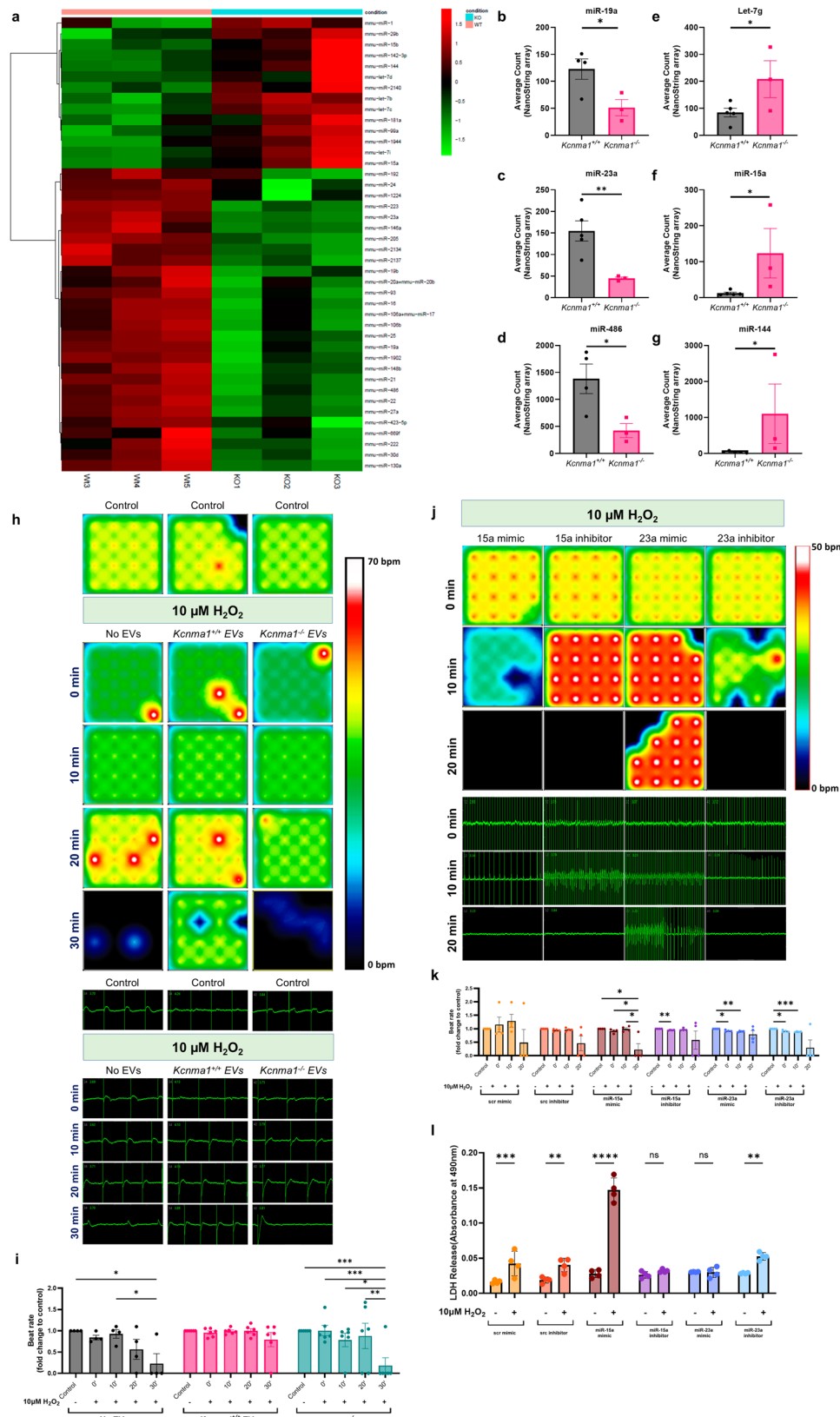

## BK$_{Ca}$ channels in EVs mediate cardioprotection from ischemia-reperfusion injury

BK channels are known to regulate cardiac physiology and activation results in cardioprotection[20–27]. We further tested the role of EVs derived from *Kcnma1*$^{+/+}$ mice in protecting the heart from ischemia-reperfusion injury. Mice were injected directly into the left ventricle

with EVs derived from *Kcnma1*$^{+/+}$ or *Kcnma1*$^{-/-}$ mice during the reperfusion. EVs isolated from *Kcnma1*$^{+/+}$ mice presented $30 \pm 10\%$ fibrosis in hearts (Fig. 4a, c). On the contrary, EVs from *Kcnma1*$^{-/-}$ mice showed increased fibrosis ($55 \pm 5\%$) in 3 independent hearts (Fig. 4b, c). Next, the cardiac function was measured after 24 h by echocardiography. The cardiac function measurements (left

**Fig. 3 | EVs isolated from *Kcnma1*⁺/⁺ mice enriched with cardioprotective miR-NAs mediate cardioprotection in hiPSC-CMs from oxidative stress induced by H₂O₂. a.** Nanostring analysis shows miR's differentially upregulated (14) and downregulated (28) in *Kcnma1*⁺/⁺ EVs compared to *Kcnma1*⁻/⁻ EVs. **b-d.** miR-19a, miR-23a, and miR-486 were upregulated in *Kcnma1*⁺/⁺ EVs compared to *Kcnma1*⁻/⁻ EVs. **e-g.** Let-7g, miR-15a, and miR-144 were down-regulated in *Kcnma1*⁺/⁺ mice as compared to *Kcnma1*⁻/⁻ EVs. **h.** Activity map of a multi-electrode array (MEA) in hiPSC-CMs showing beat frequency and field potential duration. **i.** Beat rate in hiPSC-CMs incubated with no EVs, *Kcnma1*⁺/⁺ EVs, and *Kcnma1*⁻/⁻ EVs groups. At 0, 10, 20 and 30 min after treatment with 10 μM H₂O₂. The beat rate and field potential duration were recorded at baseline and H₂O₂ treatment. **j.** Activity map of an MEA in hiPSC-CMs showing beat frequency and field potential duration after miR treatment. At 0, 10, and 20 min after treatment with 10 μM H₂O₂ beat rate and field potential duration were recorded. **k.** Beat rate in hiPSC-CMs treated with 15a mimic, 15a inhibitor, 23a mimic, and 23a inhibitor post H₂O₂ treatment. **l.** Quantification of the LDH released by scr mimic-, scr inhibitor-, miR-15a mimic-, miR-15a inhibitor-, miR-23a mimic- and miR-23a inhibitor- treated hiPSC-CMs in the presence or absence of H₂O₂. All data are presented as mean values ± SEM with the individual biological samples shown. **b-g.** Statistical analyses were carried out using a two-tailed Student's *t*-test (n = 3-4 biological repeats). **i, k and l.** Significance was determined by two-way ANOVA with Tukey's multiple-comparison test for comparison of the control baseline mean against the group with/without H₂O₂ treatment and over-time (n = 3–6 biological repeats) and *$p \leq 0.05$, **$p \leq 0.01$, ***$p \leq 0.001$, ****$p \leq 0.0001$.

ventricular ejection fraction and fractional shortening) showed a significant reduction in the cardiac function of mice after ischemia and reperfusion injury (Fig. 4d–h). However, the cardiac function showed improvement in hearts injected with EVs isolated from *Kcnma1*⁺/⁺ mice *vs. Kcnma1*⁻/⁻ mice (Fig. 4d–h). These findings confirmed the EV content, indicating that the lack of a functional BK$_{Ca}$ channel differentially affects the miRNA content.

## Discussion

Overall, our results demonstrated that EVs possess functional ion channels that are essential for maintaining their integrity and functional roles. This study focused on large-conductance K⁺ channels (BK$_{Ca}$) as the Nernst potential for K⁺ ions is the highest across the membranes. The IBTX-sensitive currents presented 300 pS conductance, which was absent in the EVs isolated from *Kcnma1*⁻/⁻ mice. We predict that during the transfer of EVs from host cells to the recipient cell, a change in K⁺ concentration will activate BK$_{Ca}$ channels. Activation of BK$_{Ca}$ channels will rapidly allow EVs to neutralize the osmotic shock without undergoing any damage. Similarly, the absence of BK$_{Ca}$ channels affects a subset of EVs, which we predict are dependent on BK$_{Ca}$ channels. The major limitation is the orientation of the BK$_{Ca}$ channel is not deciphered in EVs. Given a high concentration of Ca²⁺ in the extracellular milieu, whether the Ca²⁺ sensing domain is present in the luminal side or inside the vesicle, we anticipate that the BK$_{Ca}$ channel will rapidly open to handle the osmotic shock.

EVs found in the plasma of *Kcnma1*⁻/⁻ mice are viable and are likely to be compensated by other potassium channels to manage osmotic shock. This is demonstrated by the detection of IBTX-insensitive currents in EVs and in the planar bilayer, where we recorded smaller K⁺ conductance. In line with earlier reports[18,28], we discovered the transporters in EVs that could possibly contribute to K⁺ replenishment in EVs. The knowledge that ion channels are present and are vital for the integrity of EVs can be used to identify further and characterize other ion channels in nanovesicles. Future work focusing on the identification of ion channels and transporters present in EVs, and the measurement of precise ionic concentrations[29] in EVs will enhance our understanding of EV ionic homeostasis.

EVs present a wide heterogeneity in their sizes and content. The precise mechanism of EVs heterogeneity is not yet elucidated. However, biogenesis in the originating cell and the downstream target cell are some of the key predictors of the variability in EVs. Our findings indicated that ion channels, including BK$_{Ca}$, play a role in the heterogeneity of EVs with respect to their sizes and contents. We predict that BK$_{Ca}$ channels possibly contribute to biogenesis and the enrichment of EVs in a specific environment. The presence or absence of BK$_{Ca}$ channels can result in differential expression of miRNAs[30]. During biogenesis, the presence of BK$_{Ca}$ channels can indicate the origin of the EVs, which will define the cargo attached to the membrane or packaged. Once the nanovesicle is released, the presence and activity of ion channels, such as BK$_{Ca}$ channels, can decide its fate. The absence or inactivation of BK$_{Ca}$ channels may lead to the loss or enrichment of a specific cargo. Activating of BK$_{Ca}$ channels enables EVs to manage osmotic shock and carry charged cargo. We propose using ion channels and transporters as potential markers to select EVs with specific cargo.

The emergence of EVs as a potential therapeutic approach in protecting cells and cardiac tissue has gained prominence[31]. EVs play a crucial role in (patho)physiological processes, such as maintaining cellular balance, promoting cancer progression, and facilitating cell-cell communication through various types of nucleic acids, including mRNAs, miRNAs, ribosomal RNAs (rRNAs), and small nuclear RNAs (snRNAs)[32]. Several studies have implicated EVs in protecting the myocardium, possibly by reducing apoptosis and preserving the cardiac contractility[31,33]. We found that EVs from wild-type mice are enriched with cardioprotective miRNAs, which we further examined for their protective roles in cardiomyocytes. Since miRNAs that are known to protect the heart exhibit effects similar to those of EVs from wild-type mice, we hypothesize that BK$_{Ca}$ channels play a role in enhancing the enrichment of these cardioprotective miRNAs. In contrast, the absence of BK$_{Ca}$ channels may lead to the loss of protective EVs or disrupt the formation of EVs containing cardioprotective cargo. Furthermore, BK$_{Ca}$ channel activation is associated with cardioprotection and overall cardiac function[20,22,23,26,34–36]. The presence of functional BK$_{Ca}$ channels in EVs highlights their protective role in the adult heart and suggests their potential as therapeutic targets for cardiac and neuronal protection[34,37]. However, these studies do not rule out the role of BK$_{Ca}$ channels present in other organelles in cardioprotection.

Our discovery of functional BK$_{Ca}$ channels in EVs highlights the diversity of these nanovesicles that are actively secreted by different cell types. This finding encourages further exploration of the roles that various ion channels and transporters play in EVs, which is important for understanding how these vesicles affect cellular behavior and integrity. Identifying other functional ion channels and transporters in EVs is key to understanding their roles in regulating cargo selection and influencing the responses of recipient cells. The integrity of EVs involves not just their structure but also the functional outcomes that result from interactions between these channels and their environment. These interactions are essential for maintaining cellular and organ function, impacting processes like ion balance, signaling, and metabolism.

Additionally, examining the functional outcomes of different ion channels and transporters across various cellular and organ systems will enhance our understanding of how nanovesicles operate in physiological processes. This knowledge is important for comprehending how EVs can affect health and disease, especially in terms of cell communication and the transfer of bioactive molecules. While progress has been made in this area, the study of EVs in disease conditions is still developing. As we explore their molecular composition and functions, a broader understanding of ion transport mechanisms will be important. This comprehensive approach can help identify new biomarkers for diseases and potential therapeutic targets.

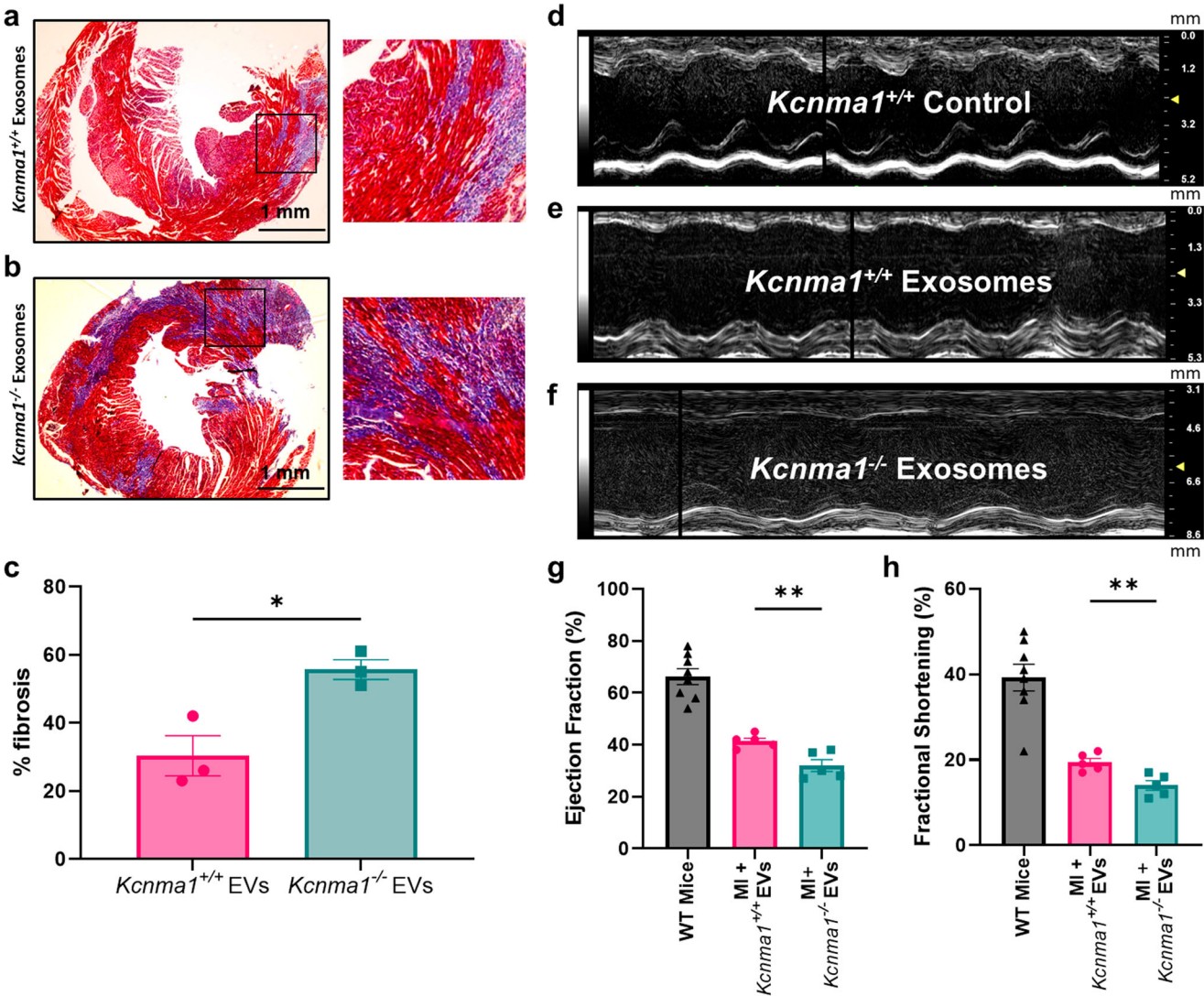

**Fig. 4 | EVs transplanted from *Kcnma1*+/+ mice improve cardiac function post-MI.** *Wild-type* mice were subjected to in vivo ischemia-reperfusion injury model. All mice underwent 45 min of LAD ligation, followed by reperfusion for 72 hrs. EVs from *Kcnma1*+/+ and *Kcnma1*−/− mice were delivered intramyocardially into the heart after reperfusion. The cardiac sections were stained with Masson Trichrome to assess cardiac fibrosis in Wild-type mice transplanted with EVs isolated from **a**. *Kcnma1*+/+ and **b**. *Kcnma1*−/−. **c** Wild-type animals injected with EVs isolated from *Kcnma1*−/− showed increased fibrosis when compared to EVs isolated from *Kcnma1*+/+ mice. Quantification of left ventricular fibrosis (%) in *Kcnma1*+/+ (blue) and *Kcnma1*−/− (red) mice (n = 3, respectively). **d** M-mode image of echocardiogram from wild-type mice without MI. **e** M-mode image of echocardiogram from wild-type mice post-MI and transplanted with EVs isolated from *Kcnma1*+/+ mice. **f** M-mode image of echocardiogram from wild-type mice post-MI and transplanted with EVs isolated from *Kcnma1*−/− mice. **g** Quantification of left ventricular ejection fraction (LVEF) of all groups of mice (sham and ligated mice injected with EVs isolated from *Kcnma1*+/+ and *Kcnma1*−/− mice). **h** Quantification of left ventricular fractional shortening (LVFS) of all groups of mice. Ligated mice showed a reduction in LVEF and LVFS as compared to sham mice (wild type). However, mice transplanted with EVs isolated from *Kcnma1*+/+ mice showed improved LVEF and LVFS as compared to EVs from *Kcnma1*−/− mice (n = 5, respectively). Significance was determined by using a two-tailed Student's *t*-test (n = 3–8 mice) with *$p \leq 0.05$, **$p \leq 0.01$.

In summary, further research into the roles of ion channels and transporters in EVs will deepen our understanding of their functions and could lead to practical applications in clinical settings. By utilizing the unique properties of EVs, we may develop new diagnostic tools and therapies that leverage their roles in cellular communication and disease processes.

## Methods

### Ethics declaration
All animal studies followed the Guide for the Care and Use of Laboratory Animals (NIH Publication, 8th Edition, 2011) and procedures were approved by the Institutional Animal Care and Use Committee of The Ohio State University under protocol 2018A00000095 and human plasma were processed after the Human Research Ethics Board Approval from University of Groningen (Lifelines project number: OV21_00315).

### Isolation and characterization of EVs by nanoparticle tracking analysis (NTA)
Blood was collected in 3.2% (*w/v*) sodium citrate from the submandibular vein of three month old male *Kcnma1*+/+ and *Kcnma1*−/− in C57BL/6NCrL background mice[38]. Blood plasma was separated by centrifugation. Plasma EVs were isolated using ExoQuick-TC® (System Biosciences, Palo Alto, CA, United States) per the manufacturer's protocol. The precipitated EVs were suspended in PBS (normalized to plasma volumes) and stored at −80 °C. To access the concentration and size distribution of isolated EVs, equal volumes of EVs in PBS were analyzed with a NanoSight NS300 (Malvern Panalytical, Malvern,

United Kingdom)[39]. Camera setting and detection threshold were kept constants for all samples for direct comparison, samples were analyzed in triplicate, and a total of at least 1000 validated tracks per sample was measured (NTA v3.3, Malvern Panalytical, Malvern, United Kingdom). Curves represent the means of triplicate measurements. To determine the size distribution in the presence of different $K^+$ concentrations, the EVs were diluted in 4 mM $K^+$, with or without NS1619 (10 μM) and IBTX (100 nM), and analyzed using Nanosight NS3000.

## $K^+$ current in lysed EV membranes

Lipid bilayers were prepared by mixing PO-phosphatidylserine (POPS), cholesterol, and 1-palmitoyl-2-oleoyl phosphatidylethanolamine (POPE) (Avanti Polar lipids, Alabaster, AL) in the molar ratio of 1:1:4 at room temperature[40–43]. Lipids were dried with nitrogen gas and resuspended in n-Decane (25 μg total lipid/μL). The lipid mixture was applied across a 0.2 mm orifice in a polystyrene cuvette, which was positioned into a black Delrin chamber partition. The cis chamber was voltage-clamped using a bilayer clamp amplifier (BC-535, Warner Instruments) connected via agar salt bridges and filtered through an 8-pole low-pass Bessel filter (Warner Instruments). For all the bilayer recordings, the lipid bilayer exhibited a capacitance of $105 \pm 10$ pF (n = 30). Bilayers were formed in 500 mM $KCH_3SO_3$ *cis vs.* 50 mM $KCH_3SO_3$ *trans* (pH 7.4). EVs were also reconstituted on Orbit mini (Nanion) in 500 mM $KCH_3SO_3$ for a quality check for electrophysiology. EVs were lysed using sonication and added to the *cis* chamber containing 1 mM DTT. The solution containing EV membranes was stirred and the $K^+$ currents were measured. 50 μM $CaCl_2$ was added to the *cis* and *trans* chamber upon initial measurements of $K^+$ currents. For the voltage *versus* current relationship, pluses were applied from −100 to +100 mV with steps of 20 mV. Iberiotoxin (IbTx) was added to the cis chamber after establishing stable baseline currents. Open probability ($P_o$) for all the recordings was obtained from recordings held at +100 mV before and after the addition of IbTx.

## Near field electrophysiology (NFE)

As reported earlier, our novel chemoelectrical sensor has shown high sensitivity towards ionic changes based on the varying charge density of the deposited polymer, geometry, and volume-to-surface area ratio during the manufacture of the polypyrrole-dodecyl benzene sulfonate [PPy(DBS)] sensor[12–14,44]. The sensing paradigm referred to as near-field electrophysiology (NFE), has been used to measure $BK_{Ca}$ current in intact EVs. The measured currents are converted to charge displaced as a function of time and used to calculate the following (i) intracellular concentration, (ii) transmembrane ionic currents in response to channel blocker, membrane lysis, and externally imposed chemoelectrical gradient. The chronoamperometric detection of extracellular $K^+$ concentration is based on the redox activity of the working electrode (WE) with respect to an integrated counter/reference (CE/RE) electrode. The WE consists of a Pt-wire (25 μm diameter) and electropolymerized with PPy(DBS) and the CE/RE containing chlorinated (250 μm diameter) Ag wire. The WE and CE/RE are arranged in concentric quartz and borosilicate capillary tubes that maintain a constant distance between the electrodes, as shown in Fig. 1. As the PPy(DBS) membrane in the WE recognizes a switch between the oxidized and reduced states (0 V and −800 mV) at a frequency of 10–20 Hz[12,15], ion transport into and out of the WE depends on the extracellular concentration. The sensor was equilibrated in a bath buffer containing EVs, and the continuous regulation of the chemoelectrical gradient by active ion transport mediated by ion channels present on the EV membranes influences the measurement of $K^+$ currents.

## Cryo-TEM of isolated EVs

Isolated EVs were sent for Cryo-TEM processing and imaging[45]. Briefly, an FEI Vitrobot Mark IV plunge freezer (Thermo Scientific, Waltham, MA, United States), set at room temperature and ~95% humidity, was used to prepare vitrified cryo-TEM specimens from the aqueous samples. About 2.5 μL of the solution was applied to a TEM grid coated with lacey carbon film. After blotting using two filter papers, the grid was plunge-frozen in liquid ethane[39]. The vitrified specimen was mounted onto a Gatan 626. The DH cryo-TEM holder was transferred into a FEI Tecnai F20 TEM equipped with a Gatan twin blade retractable anti-contaminator. The cryo-TEM observation was carried out at −174 °C.

## Western blot analysis

Isolated EVs from the plasma of three month old male $Kcnma1^{+/+}$ and $Kcnma1^{-/-}$ mice were lysed with modified RIPA buffer (Tris-HCl 50 mM, NaCl 150 mM, EDTA-$Na_2$ 1 mM, EGTA-$Na_4$ 1 mM, $Na_3VO_4$ 1 mM, NaF 1 mM, Nonidet P-40 1% (v/v) Na-deoxycholate 0.5% (w/v), and SDS 0.1% (w/v), pH 7.4) containing protease inhibitor (1 tablet/50 mL; Roche) and PhosSTOP™ (1 tablet/10 mL; Roche). The samples were flash-frozen in liquid nitrogen and incubated for 1 h at 4 °C. Afterward, they were centrifuged at $10,000 \times g$ for 20 min, and the supernatants were collected as lysates. Proteins (50 μg per lane) were separated using 4–20% SDS-PAGE and transferred onto nitrocellulose membranes. Loading accuracy was verified with Ponceau S staining. The membranes were then blocked with Intercept® blocking buffer for 1 h at room temperature, followed by overnight incubation with anti-$BK_{Ca}$[23] (Alomone labs, #APC21), anti-Na-K-ATPase (Abcam, #ab7671), anti-TOMM20 (Thermo Fisher Scientific, #MA5-32148), anti-ALIX (Cell Signaling Technology, #2171S), anti-GAPDH (Cell Signaling Technology, #2118), anti-CD-81 (System Biosciences, #EXOAB-CD81A-1), anti-E-Cadherin (Thermo Fisher Scientific, #14-3249-82). Membranes were washed thrice with 1X Tris-buffered saline containing Tween-20 and incubated with 0.01 μg/mL secondary antibody (IR-dye 800 goat anti-rabbit IgG; LI-COR Biosciences; 925-68071) for 60 min at room temperature. After extensive washing, membranes were visualized using BioRad ChemiDoc MP.

## Immunocytochemistry

Human-induced pluripotent stem cells derived cardiomyocytes (hiPSC-CMs) were incubated PKH67 labeled plasma-derived EVs isolated from $Kcnma1^{+/+}$ and $Kcnma1^{-/-}$ mice for 8 hrs. After incubation, hiPSC-CM were stained with wheat germ agglutinin (WGA) at 37 °C on ice for 60 min and/or mitotracker for 10 min at 37 °C. The cells were then fixed with 4% (w/v) PFA and permeabilized with 0.5% (v/v) Triton-X-100. HiPSC-CM were incubated with anti-$BK_{Ca}$ antibodies[23] (Alomone labs, APC21) overnight at 4 °C. Secondary antibodies conjugated with anti-rabbit Atto-647N (Sigma-Aldrich; 40839) were added for 60 min at room temperature. To label nuclei, DAPI was added (1:10,000 dilution) to the wash solution. Coverslips were mounted with ProLong Gold Antifade Mountant (Thermo Fisher Scientific, P36934). Cells were imaged with Nikon A1R high-resolution confocal microscopy. The colocalization index was determined through protein proximity index analysis[46], with image filtering done via custom-built software[23,47].

## Multielectrode array (MEA)

Human-induced pluripotent stem cell-derived cardiomyocytes (hiPSC-CMs) were plated onto CytoView MEA-24 well plates (Axion BioSystems, Inc.), coated with fibronectin (50 μg/mL), at a density of 40,000 cells per well. Experiments were performed and data was collected 72–96 h after plating using the Maestro Edge MEA platform (Axion BioSystems, Inc.). Voltage recordings were taken simultaneously from 16 electrodes per well and sampled at 50 μV. For data analysis, cardiac field potential duration (FPD) and beat period were plotted as fold changes, comparing treatment groups ($H_2O_2$, $Kcnma1^{+/+}$, and $Kcnma1^{-/-}$) to their respective baseline measurements. The CiPA Analysis Tool software (Axion BioSystems, Inc.) was used for automated detection and graphical representation of the data.

## Lactate Dehydrogenase (LDH) Assay

To measure the cytotoxicity, LDH assay was performed using the in vitro toxicology assay kit, lactic dehydrogenase based (Cat# TOX7-1KT, MilliporeSigma, Milwaukee, WI, United States). The conditioned culture medium was collected post-treatment and centrifuged at 3000 g for 5 min to remove cell debri. The LDH levels in the medium was assessed per the manufacturer's instructions. The background and primary absorbance of the plate were measured on a spectrophotometer (Tecan Infinite F200 PRO) at 690 and 490 nm, respectively. The assay was performed in quadruplicate (n = 4) and the data obtained was analyzed by subtracting background absorbance from primary absorbance.

## miRNA analysis

miRNA content of EVs was assessed in triplicate samples according to manufacturers' protocol (NanoString Technologies, Inc. Seattle, USA) by using nCounter Human v3 miRNA Expression Assay Kit (Cat# GXA-MIR3-12)[48]. Briefly, 100 ng of total RNA was annealed with multiplexed DNA tags (miR-tag) and bridges target specifics. Mature miRNAs were ligated to specific miR-tags and excess tags were removed via enzyme clean-up. The tagged miRNA product was diluted and incubated overnight with Reported Probes in hybridization buffer and Capture probes. Excess probes were removed using two-step magnetic bead-based purification on an automated fluidic handling system (nCounter Prep Station) and target/probe complexes were immobilized on the cartridge for data collection. The nCounter Digital Analyzer collected the data by taking images of immobilized fluorescent reporters in the sample cartridge with a CCD.

## miRNA transfection protocol

To study the effect of miRNAs on hiPSC-CMs function, 40,000 cells were seeded per well on Cytoview 24-well plate (Axion Biosystems). The cells were cultured for 5 days in iCell cardiomyocyte maintenance medium. For modulation of miRNA expression, the cells were transfected with 25 nM of miRCURY LNA miRNA mimics or 50 nM of miRCURY LNA miRNA inhibitors for hsa-miR-15a-5p and hsa-miR-23a-5p (Qiagen) using Lipofectamine RNAiMAX (Thermofisher) according to the manufacturer's instructions. After 16 h the cells were treated with $H_2O_2$ and MEA analysis was performed as described above.

## Induction of myocardial infarction (MI)

The ischemia-reperfusion injury was performed on three month old C57BL/6NCrL mice with slight modification[23,49]. The left anterior descending (LAD) coronary artery was ligated for 45 min followed by reperfusion for 72 h. In brief, a small incision was made over the left chest, followed by the placement of a purse-string suture. After dissection, the pectoral and minor muscle were retracted to expose the 4th intercostal space, where a small hole was created. The heart was gently "popped out" through this opening, and the left descending coronary artery (LCA) was ligated using a 6-0 silk suture, 2–3 mm from its origin, forming a slipknot. After ligation, the heart was returned to the thoracic cavity, and the air was manually removed before closing the muscles and skin. The internal end of the slipknot was cut short, while the external end, about 0.8 cm long, was left outside the chest. After 45 min of ischemia, reperfusion was initiated by gently pulling the external suture to release the slipknot. After 15 mins of reperfusion, EVs (150 μg/30 μL total volume) were injected intramyocardially into three peri-infarct areas (10 μL/area) using a 29-gauge needle (Hamilton Gastight Syringe, Model 1702). The chest cavity was closed, and the animals were allowed to recover.

## Statistical analysis

All data points are indicated as individual biological replicates in the graphs and represented as mean ± SEM. All the data points in the current *versus* voltage plot are represented as mean ± SD. The statistical analysis between the two groups were conducted using an unpaired two-tailed *t*-test. Two-way ANOVA followed by Tukey's multiple-comparison test was used to compare across different treatments, pre-/post- $H_2O_2$ treatments. No outliers were removed from the analysis. A *p*-value of ≤0.05 (*) was considered significant for all tests, where **$p ≤ 0.01$, ***$p ≤ 0.001$, ****$p ≤ 0.0001$. Statistical analysis were performed using GraphPad Prism Ver 10.3.1 (GraphPad Software). Images were filtered by custom-built software[23,47] and curated using ImageJ (National Institutes of Health, Bethesda, MD, USA). The Western blot quantification was performed using ImageJ.

## Reporting summary

Further information on research design is available in the Nature Portfolio Reporting Summary linked to this article.

## Data availability

All data needed to evaluate the conclusions in the paper are present in the paper and/or the Supplementary Materials. Source data are provided with this paper.

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

## Acknowledgements

We thank the cardiac arrhythmia group at the Ohio State University for the helpful discussions. We also would like to thank Dr. Vishnu Baba Sundaresan from the Department of Mechanical and Aerospace Engineering, The Ohio State University, Columbus, OH, for assisting with the Near Field Electrophysiology experiments. Prof. Seena Ajit (Drexel University College of Medicine), and Prof. Raj Kishore (Temple University) for helpful discussions on extracellular vesicles, and Dr. Ramakant Lawaniya (Hygiena International) for EV ATP discussions. This work was supported by The Ohio State University President's Predoctoral Fellowship, The Ohio State University Department of Physiology and Cell Biology Margaret T. Nishikawara Merit Scholarship Endowment in Physiology and The Ohio State University Graduate School's Alumni Grants for Graduate Research and Scholarship Program to S.S. American Heart Association's

Postdoctoral Fellowship (916599) to D.S. National Heart, Lung, Blood Institute (NHLBI) (HL157453) and National Institute of Arthritis and Musculoskeletal and Skin Diseases (AR080946) to M.K. American Heart Association–Transformational Project Award (AHA-TPA) (972077) to S.G.R. National Center for Advancing Translational Sciences (TR004344), NHLBI (HL133050 and HL157453), and AHA-TPA (965301) to H.S.

## Author contributions

S.S., D.S., M.K., and H.S. conceptualized the project. S.S. and H.S. wrote and revised the manuscript. S.S., D.S., P.E., J.D., K.S., D.G., M.D., J.W., S.G.R., S.V.P., D.R.H., T.C., P.S.A., A.M.D., L.Y., and H.S. performed experiments. S.S., D.S., and H.S. analyzed the data. L.Y. performed biostatistical analysis.

## Competing interests

The authors declare no competing interests.
