## [Transparent Peer Review file · Nature Communications]

Functional large-conductance calcium and voltage-gated potassium channels in extracellular vesicles act as gatekeepers of structural and functional integrity

Corresponding Author: Professor Harpreet Singh

Version 0:

Reviewer comments:

Reviewer #1

(Remarks to the Author)

The manuscript by the group of Harpreet Singh reports on the presence of calcium activated BKCa channels in extracellular vesicles (EV) and on its role in the regulation of EV size and number, miRNA loading of EVs and on the cardioprotective role of EV-located BKCa.

While the findings are interesting and might be of relevance, the work is rather descriptive and mechanistic insights are mostly lacking, therefore leaving the reader with many open questions.

The major issues that should be addressed before publication are the following:

The authors should provide biochemical evidence for the purity of the EV they isolated regarding the eventual contamination by various cell compartments/membranes where BKCa is located. Since EVs originate from intracellular membranes, the question is whether the BKCa found in EVs arise e.g. from mitochondria or other intracellular membrane or it is present in the EV preparation due to PM contamination.

The link between the presence of such channel in EVs and of the different types of miRNAs is not clear from mechanistic point of view. The authors mention that BKCa was shown to regulate protein content in EVs, but they give no rationale for testing the miRNA-BKCa link and no explanation is provided regarding the selective up-or –downregulation of certain groups of miRNAs depending on the absence of BKCa in EVs.

The authors report on an increased number of EVs from BKCa KO animals with respect to WT animals, but do not discuss this finding. Given that BKCa according to the results presented in this paper regulates EVs integrity, the slightly decreased size might be logical, but the increased EV number is not easy to understand.

Data are presented with different graph styles in the figures. Please unify, showing graphs with single data points as in Supplementary Figure 4G.

In most cases the number of biological and technical replicates is not reported in the main text or in the legend. This must be fixed.

The quality of the electrophysiological data is low, the current traces are noisy. In suppl. Fig 2B, it seems that the experiment was done only once, there is no SD reported on the graph.

The authors write: "EVs are endosomal in origin and contain 9,769 proteins, 3,408 mRNA, 2,838 miRNA, and 1,116 lipids". Please specify the system for which these numbers refer to.

"Our in silico analysis indicated that there are 72 unique ion channels and 107 transporters or exchangers present in EVs (Supplementary table 1)." There is no mention of the source and methods used. Please specify the organism, the method and database used for this analysis.

"In physiology, the ions with the highest gradient across the plasma membrane are [K+] ions." This statement does not take into account calcium ion gradient that is higher than that of K+. The Nernst potential for calcium is $V_{Ca} = +137.04$ mV against V_K of -96.81 mV.

Please write always the same way KMeSO4 solution and specify why it is useful for these experiments.

Figure 1H: it would be useful to enhance lbtX concentration and show complete block.

Supplementary Figure 4 shows relevant in vivo experiments. On my opinion these data could be included in the main text.

“As compared to PBS, the mean size of EVs decreased from 177.43 ± 2.27 nm to 161.36 ± 9.33 nm in 4 mM K⁺ but EVs size increased to 217.06 ± 12.59 nm in 145 mM K⁺”

It is not clear whether the authors refer to extra or intraEV concentrations.

Please specify what is WGA and what is PKH67, why is it used as EV marker.

Figure 3H: Please specify if the H₂O₂ concentration used is physiologically relevant.

“Our finding, in agreement with exosomal content, where we discovered that an absence of functional BKCa channel increases deleterious miRs and decreases cardioprotective miRs.” Please fix this sentence.

Supplementary figure 3: it would be useful to repeat the Western blot using a more specific antibody that does not give rise to the 100 kDa unspecific band.

Reviewer #2

(Remarks to the Author)

In this study, the authors first investigated if a K channel (BKCa) is present and functional on Extracellular Vesicles, and showed that the channel maintains the integrity (size) of the vesicles. Then they took advantage of BKCa null mice and compared the miRNA content of plasma EV emanating from BKCa null mice or WT mice. They identified miRNAs that were down or upregulated. They also showed that EV containing BKCa exert a cardioprotective function. Treatment with miRNAs enriched in BKCa positive EV phenocopies those treatment with EV.

This study is articulated in two parts. First the demonstration of the presence of the BKCa in EVs, and a validation of its functionality, then a correlation between BKCa, miRNA content and cardioprotective effect.

Each observation is interesting and important on its own and deserved further characterization.

Below are some of my major concerns about the results and the support of the claim

. The EV characterization by dot blot (Fig1A) is not convincing. It is impossible to clearly distinguish the positivity for candidates such as Alix (that should be positive) or Gm130 (that is expected to be negative). Since the authors master western blot, they should assess the presence or absence of each proposed marker in the isolated EVs and compare it with cell lysate (equivalent amount of protein). This will address if the protein is present AND enriched within EV. In general, controls (negative and positive are lacking to fully validate the results)

. The authors tested the “size of EVs” in various buffers with high or low K⁺ concentration, by NTA and refer it to EV integrity. EV integrity also corresponds to membrane integrity (or rupture). This should be complemented by protease protection assay to test if the vesicle are intact (an EV cargo such as Alix is protected) or if there is membrane rupture (the EV cargo is degraded) in all tested buffer. The authors mentioned that cell depleted of BKCa channel release smaller vesicle in higher number. This should be discussed.

. Figure 2 and Supplementary Fig3 aim at assessing and validating the absence of BKCa in EV and donor cells. It is very confusing because while the text mentions a highly specific antibody, the two shown immunoblots are different; one shows a single band at 125 kDa whereas the other one shows a ladder (Fig2). Again, as for figure 1 this experiment needs to include rigorously and systematically all the controls side by side. The authors should load equal protein amount of CL from WT and BKCa null cells and EVs to enable rigorous comparison.

.The author claim that the uptake is not affected by the presence or absence of BKCa. Fig2J-U should be quantified to support this claim. Ideally this should be confirmed with an independent approach (FACS?). It is not clear why many PKH26 positive EVs are not BKCa positive in Fig 2K and L. The colocalization is almost perfect in Fig2B and C. the authors need to explain this apparent discrepancy.

. Finally my most important concern is the demonstration of the miRNA effect thru EVs. The authors roughly showed that treatment with miRNA15 phenocopy the treatment with BKCa positive EV. The protocol for miRNA treatment is not clear. The author used the term “directly incubated cell with miRNA 15...”. What does it mean? miRNA were added in the culture media? If yes how do they enter the cell? Is it microinjection? What is the concentration of the used miRNA (the initial one and the one inside the cell) ? is it the same that the one in the EV? Are the miRNA 15 inside the EV or at the surface ? Dose-response and kinetics should be performed to rigorously measure the miRNA delivery and judge if the the miRNA treatment is comparable with EV treatment at least in term of miRNA concentration. At best, this part establishes indirect correlation but does not fully support the direct causality between BKca-miRNA15 nad cardioprotective effect. At the very least the authors should town down the conclusion and the interpretation.

-The material and method section was in the supplementary files, it should be in the main text.

Reviewer #3

(Remarks to the Author)

In this study, the authors postulate that BKCa channels may be present in EVs, and play a functional role. They use a near field electrical potential measurement to show that incorporation of EVs elicits a larger current response that is partly suppressed by the BK channel toxin iberiotoxin. They also observe channel activity resembling BK in bilayers containing

reconstituted EVs. The size and number of EVs appeared to differ in preparations from KCNMA1^{+/+} mice compared to KCNMA1^{-/-} mice. Incorporation of EVs into hiPSC-derived cardiomyocytes showed that they were located intracellularly. miRNA profiles were measured in EVs from ^{+/+} or ^{-/-} mice and several species were significantly altered. The authors go on to speculate that this difference in profiles might affect cardioprotection by said EVs, and some differences in the beat rate response of hiPSC-CMs to H₂O₂ were observed.

BKCa channels have been well documented in intracellular organelles, such as lysosomes (<https://doi.org/10.1083/jcb.201612123>), which are incorporated into EVs (<https://doi.org/10.3390/ijms232315317>). Hence, it is not surprising that they would be present in EVs, although it is unclear what orientation they would have with respect to the outer membrane of the EV and what impact they would have on transport. Can the organelle of origin of the BK channels be determined? Are the EVs positive for lysosomal markers?

Fig. 1: It is not clear that EV membranes would maintain polarization in extracellular solutions, particularly since BKCa channels, if present, should be highly activated in the high Ca²⁺ external environment. Moreover, it is debatable as to whether ion gradients could be maintained by ion pumps in the absence of ATP. Was the current transient recorded in the presence of EVs abrogated in the presence of a K⁺ ionophore or NaKATPase inhibitor? The evidence underpinning the validity of the NFE method is not provided.

In Fig. 1F, the bilayer experiments show a channel that does not seem to be very K⁺ selective. With the K⁺ gradient mentioned (described as 500:50 mM in the legend, but 150:50 in the results section), the reversal potential should be in the neighborhood of -60 mV but it is closer to -20mV in panel 1F. Does this reversal point shift as expected when the K⁺ gradient is changed? Were these experiments repeated more than once?

Fig. 3: Beat rate is not a direct measure of cellular injury, as any change to the resting membrane potential or pacemaker currents could alter this parameter. Indeed, activation of K⁺ channels would also affect this parameter. Thus, the changes shown in Fig. 3H-J are of uncertain significance. Moreover, the assumption that altered miRNA profile has anything to do with the protection in EV-treated cells is not proven in this study. hiPSC-derived myocytes are usually quite tolerant of low levels of H₂O₂ (i.e. 10uM). Some additional measure of cell injury should be measured, for example, LDH release into the medium.

Fig. 4. Fibrosis is measured from random sections post-MI. These measurements can be highly variable and should take into account multiple sections taken systematically. The lower panel of 4F shows echocardiograms from *Kcnma1*^{-/-} mice, but the images are very strange, with both walls moving in the same direction during the heartbeat. Is this cardiac dyssynchrony? How can one estimate fractional shortening reliably?

Version 1:

Reviewer comments:

Reviewer #1

(Remarks to the Author)

In the revised version the authors addressed most of my comments. Importantly, they performed the requested Western blots and surprisingly found no contamination of the EVs by mitochondrial membranes. However, in Suppl. Table 1. All 3 isoforms of VDAC as well as the mitochondrial calcium uniporter (listed among transporters, please change), known to be specifically located only in mitochondrial outer and inner membranes, respectively, are found. How can this be explained? In Suppl. Fig 2 the authors forgot to add the number of experiments in the legend. Please correct.

Although the authors answered most of my questions, my feeling is that they did not properly address some of the issues raised by the other reviewers, specifically reviewer 3. For example, the question of the channel orientation and the calcium-induced high activity remain open.

Reviewer #2

(Remarks to the Author)

the authors have satisfyingly address all my comments.

Reviewer #3

(Remarks to the Author)

The authors have responded to the comments with some additional data to improve the manuscript.

Specifically: 1) They looked for markers of lysosomes and mitochondria and did not find any, indicating that the origin of the Bk channel was not likely to be from these organelles; 2) they determined that ATP was present in the EVs. This did not exactly answer the question about the origin of polarization of the EVs (the effects of a Na pump inhibitor or K ionophore might have been more informative from a functional point of view); 3) They did some hand waving about K selectivity, but did not vary the gradient to see if the reversal potential is shifted with the K gradient; 4) They added LDH measurements for the miRs, which is helpful; and 5) They found better echo recordings, which is an improvement.

In general, the reviewers have been responsive, although the precise role of the ion channels and transporters in EV ion homeostasis is somewhat unclear.

Version 2:

Reviewer comments:

Reviewer #1

(Remarks to the Author)

The authors provided an explanation to my questions.

Dear Reviewer,

We highly appreciate your suggestions and comments. We thank you for understanding that in the current manuscript we “sought to identify and characterize a functional potassium channel in extracellular vesicle membranes which is crucial for their survival and regulation of their cargo”. Therefore, the focus of the manuscript is exclusively on the existence and functional characterization of BK channels in EVs. We have addressed and incorporated your criticisms and suggestions, respectively in the manuscript, performed additional experiments, and also provided a detailed response here in blue;

Reviewer #1 (Comments to the Author):

The authors should provide biochemical evidence for the purity of the EV they isolated regarding the eventual contamination by various cell compartments/membranes where BKCa is located. Since EVs originate from intracellular membranes, the question is whether the BKCa found in EVs arise e.g. from mitochondria or other intracellular membrane, or it is present in the EV preparation due to PM contamination.

We agree with the reviewer’s comment on the purity of the EVs, which are devoid of plasma membranes and various cell compartment contamination. Hence, we have incorporated the data to include the Western blot data in Figure 1, suggesting the EV purification does not have E-cadherin (plasma membrane) or GRP78 (endoplasmic reticulum), LAMP-1 (Lysosome) and TOMM20 (mitochondrial) contamination but are positive for EV markers such as CD81, GAPDH, and ALIX. In the follow-up work of this manuscript, we plan to characterize the EV transcript of the BK_{Ca} channel arising from ER (BK_{Ca} SV27), mitochondria (BK_{Ca} DEC), or plasma membrane (BK_{Ca} FL). However, these characterizations will be published in the follow-up manuscript.

The link between the presence of such channel in EVs and of the different types of miRNAs is not clear from mechanistic point of view. The authors mention that BKCa was shown to regulate protein content in EVs, but they give no rationale for testing the miRNA-BKCa link and no explanation is provided regarding the selective up-or –downregulation of certain groups of miRNAs depending on the absence of BKCa in EVs

Activation of BK_{Ca} channels has been implicated in cardioprotection and cardiac function PMID: 29438488, 23754429, 30746365. Since we found the presence of functional BK channels in the EVs, we wanted to assess if it impacts the content of these EVs. It is well known that EVs mediate cellular communication through the transfer of macromolecules encompassing nucleic acids, proteins, miRNAs, and lipids. Our study focused on understanding the differences in miRNA content in *Kcnma1^{+/+}* and *Kcnma1^{-/-}* EVs. Consistent with our hypothesis, our data showed an enrichment of cardioprotective miRNAs in *the Kcnma1^{+/+}* EVs. However, whether functional BKCa channels directly impact miRNA packaging will have to be investigated in future studies. We have incorporated the rationale for investigating the miRNA content discussion section to include the rationale for looking into miRNAs in the manuscript.

The authors report on an increased number of EVs from BKCa KO animals with respect to WT animals, but do not discuss this finding. Given that BKCa according to the results presented in this paper regulates EVs integrity, the slightly decreased size might be logical, but the increased EV number is not easy to understand.

We agree with the reviewer's concern about the increased number of EVs observed in global *Kcnma1*^{-/-} mice. One of the possible explanations is the involvement of BK_{Ca} channels in the biogenesis of the exosomes. The alternative explanation is similar to lysosomes; EVs might also undergo an increased fission process (PMID: 31171420). Though feasible, there are no studies indicating EVs undergoing the fission process.

The quality of the electrophysiological data is low, the current traces are noisy. In suppl. Fig 2B, it seems that the experiment was done only once, there is no SD reported on the graph.

The reviewer has a valid concern regarding the traces obtained in the supplementary figure 2B. We have repeated the experiments to increase the 'n' number and have plotted the IV with standard deviation instead of the standard error of mean (error bars were present but too small). Since the currents are smaller, the traces seem to be noisy as compared to the wild type.

The authors write: "EVs are endosomal in origin and contain 9,769 proteins, 3,408 mRNA, 2,838 miRNA, and 1,116 lipids". Please specify the system for which these numbers refer to.

The reviewer's curiosity about the precise number of contents within the EVs is highly appreciated. These precise numbers have been mined through the available online database Exocarta, a collection of exosomal cargo. This database consists of data collected from vertebrates and invertebrates using mass spectrometry, NanoString, Western blotting, and high-throughput OMICS techniques (PMID: 33969112).

<http://exocarta.org/browse>

"Our in silico analysis indicated that there are 72 unique ion channels and 107 transporters or exchangers present in EVs (Supplementary table 1)." There is no mention of the source and methods used. Please specify the organism, the method and database used for this analysis

We have included the source and method of data collection in the manuscript.

"In physiology, the ions with the highest gradient across the plasma membrane are [K⁺] ions." This statement does not take into account calcium ion gradient that is higher than that of K⁺. The Nernst potential for calcium is $V_{Ca} = +137.04$ mV against V_K of -96.81 mV.

We agree with the reviewer that Ca²⁺ ions have the highest Nernst potential. However, our study focuses on the univalent cation K⁺; hence, we have rephrased the statement.

Perhaps our study will trigger future studies in Ca^{2+} , Na^+ , Cl^- and other ion channels as well as transporters.

Please write always the same way KMeSO_4 solution and specify why it is useful for these experiments.

We have incorporated the reviewer's suggestion in the manuscript. KCH_3SO_3 in a solution dissociates as K^+ and CH_3SO_3^- , where the conducting ion is that of K^+ . Since no other ions are present, and the anion is too large to permeate, this is an appropriate solution to record K^+ currents specifically.

Figure 1H: it would be useful to enhance IbtX concentration and show complete block.

The IC_{50} of IBTX is 2nM. We have used ~50-fold higher concentration above the IC_{50} . We did observe a significant block. As illustrated in other manuscripts (PMID: 11487613, 19363492, 15071098) we have obtained a nearly complete block of Po.

The reviewer recommended increasing the concentration of iberiotoxin to obtain a complete block; however, the presence of residual K^+ currents are potentially mediated by other K^+ channels, as observed in *Kcnma1*^{-/-} exosomes.

Supplementary Figure 4 shows relevant in vivo experiments. On my opinion these data could be included in the main text.

Our focus is to show that ion channels exist in EVs, and physiological data was kept in a supplementary file. Since the reviewer has requested, we have moved the supplementary figure 4 in the main text.

"As compared to PBS, the mean size of EVs decreased from 177.43 ± 2.27 nm to 161.36 ± 9.33 nm in 4 mM K^+ but EVs size increased to 217.06 ± 12.59 nm in 145 mM K^+ ". It is not clear whether the authors refer to extra or intra EV concentrations.

We agree with the reviewer's comment and have clarified the intracellular and extracellular K^+ concentration in the manuscript.

Please specify what is WGA and what is PKH67, why is it used as EV marker.

We agree with reviewer's concern with the PKH67 as an EV marker. PKH67 dyes consist of a long lipophilic tail that gets inserted into the lipid bilayer structure and emits fluorescence outside the lipid membranes. Wheat Germ Agglutinin (WGA) is a lectin-based fluorescent molecule that binds to glycoproteins in the cell membrane. In EV biology, researchers have interchangeably used the PKH and WGA dyes for labeling EVs as 'standard control' (PMID: 37588627).

Figure 3H: Please specify if the H_2O_2 concentration used is physiologically relevant

In physiology, the concentration of H_2O_2 is predicted to be ~35 μM in human blood plasma (PMID: 11108833). In the study by Fiedler et al., where they induced oxidative stress

using H₂O₂ on commercially procured hiPSC-CMs (similar to our study) at a dose-dependent concentration from 5–500 μM (PMID: 30853557)

“Our finding, in agreement with exosomal content, where we discovered that an absence of functional BKCa channel increases deleterious miRs and decreases cardioprotective miRs.” Please fix this sentence.

We have reworded the statement in the manuscript.

Supplementary figure 3: it would be useful to repeat the Western blot using a more specific antibody that does not give rise to the 100 kDa unspecific band.

We completely agree with the reviewer’s concern with non-specific bands. However, we have performed extensive characterization of the BK_{Ca} antibody (Alomone labs, APC21) and have observed the second band even in our *Kcnma1*^{-/-} mice.

Reviewer #2 (Comments to the Author):

The EV characterization by dot blot (Fig1A) is not convincing. It is impossible to clearly distinguish the positivity for candidates such as Alix (that should be positive) or Gm130 (that is expected to be negative). Since the authors master western blot, they should assess the presence or absence of each proposed marker in the isolated EVs and compare it with cell lysate (equivalent amount of protein). This will address if the protein is present AND enriched within EV. In general, controls (negative and positive are lacking to fully validate the results)

We have modified Fig. 2A to include controls and markers for the EV purification to address the reviewer's concern.

The authors tested the "size of EVs" in various buffers with high or low K⁺ concentration, by NTA and refer it to EV integrity. EV integrity also corresponds to membrane integrity (or rupture). This should be complemented by protease protection assay to test if the vesicle are intact (an EV cargo such as Alix is protected) or if there is membrane rupture (the EV cargo is degraded) in all tested buffer. The authors mentioned that cell depleted of BKCa channel release smaller vesicle in higher number. This should be discussed.

We have included videos of EVs in the supplemental file to show the integrity of vesicles. This is in addition to EV electron micrographs provided in Figure 1. We have also expanded the discussion on the role of BK in the size and number of EVs.

Figure 2 and Supplementary FiG3 aim at assessing and validating the absence of BKCA in EV and donor cells. It is very confusing because while the text mentions a highly specific antibody, the two shown immunoblots are different; one shows a single band at 125 kDa whereas the other one shows a ladder (Fig2). Again, as for figure 1 this experiment needs to include rigorously and systematically all the controls side by side. The authors should load equal protein amount of CL from WT and BKCa null cells and EVs to enable rigorous comparison.

The ~100 kDa non-specific band is always present in the samples and in fact, is at higher intensity in null mutant samples. We have published this information in several papers (PMIDs: 23754429, 35393410, FCVS). We have modified Figure 2A as suggested by the reviewer. The antibody is highly specific for imaging but does give an additional band at 100 kDa and in fact occasionally at 55 kDa as published here PMID: 12411707.

The author claim that the uptake is not affected by the presence or absence of BKCa. Fig2J-U should be quantified to support this claim. Ideally this should be confirmed with an independent approach (FACS?). It is not clear why many PKH26 positive EVs are not BKCa positive in Fig 2K and L. The colocalization is almost perfect in Fig2B and C. the authors need to explain this apparent discrepancy.

Reviewer Figure 1: Uptake of WT EVs by CMs. The hiPSC-CM were treated with PKH26 labeled EVs for 12 hrs. The cells were dissociated into single cells. The cells were fixed in 4% PFA for 15 mins and permeabilized with 0.2% triton X-100 for 5 mins. The cell pellet was resuspended in 1% BSA solution for blocking for 1 hr on ice. Further, cells were washed and resuspended in PBS for analysis. Flow cytometry experiments were performed using BD LSR Fortessa system the data was analyzed by using the BD FACSDiva software.

We have quantified the data to show that EVs isolated from wild-type localize deliver BK channels and EVs derived from *Kcnma1*^{-/-} can still be loaded to cells but they lack BK signal. Therefore, the uptake mechanism is not dependent on the expression of BK channels. We have also performed FACS and included the data in the supplementary figure, as requested by the reviewer. We have also quantified the colocalization of BK to PKH-26 for 2B and 2C. We further performed the FACS experiment to confirm the uptake of exosomes by hiPSC-CMs (included here)

Finally my most important concern is the demonstration of the miRNA effect thru EVs. The authors roughly showed that treatment with miRNA15 phenocopy the treatment with BKCA positive EV. The protocol for miRNA treatment is not clear.

We have clarified and added a detailed protocol. We apologize for the confusion.

The author used the term “directly incubated cell with miRNA 15...”. What does it mean? miRNA were added in the culture media? If yes how do they enter the cell? Is it microinjection? What is the concentration of the used miRNA (the initial one and the one inside the cell)? is it the same that the one in the EV? Are the miRNA 15 inside the EV or at the surface? Dose-response and kinetics should be performed to rigorously measure the miRNA delivery and judge if the the miRNA treatment is comparable with EV treatment at least in term of miRNA concentration. At best, this part establishes indirect correlation but does not fully support the direct causality between BKca-miRNA15 and cardioprotective effect. At the very least the authors should town down the conclusion and the interpretation.

We have elaborated the methodology used for delivering miRNA mimics and inhibitors. We performed dose-response of miRNA delivery using the FAM dye-labeled has-miR-15a-5p mimic and inhibitor at 10nM, and 25nM concentrations. Furthermore, we evaluated the expression of miRs using RT-qPCR, where overexpression of miR-23a in hiPSC-CM increased the expression by 2.2 folds and overexpression of miR-15a increased by ~600 folds and the complete inhibition of miR-15a (undetected). These results validate the dose-response efficiency of the mimics and inhibitors transfection in hiPSC-CM. We have also modified the discussion to tone down the conclusion and interpretation.

Reviewer Figure 2. Dose-response kinetics of overexpression of miRs mimic and inhibitors in hiPSC-CMs. HiPSC-CMs were transfected with 10 nM and 25 nM of miR mimic or inhibitor and cultured for 16 hours. **A)** Fluorescence images showing expression FAM labeled hsa-miR-15a-5p inhibitor and mimic in hiPSC-CM. Expression levels of **B)** hsa-miR-15a-5p and **C)** hsa-miR-23a-3p analyzed under each treatment from hiPSC-CMs.

Reviewer #3 (Comments to the Author):

BKCa channels have been well documented in intracellular organelles, such as lysosomes (<https://doi.org/10.1083/jcb.201612123>), which are incorporated into EVs (<https://doi.org/10.3390/ijms232315317>). Hence, it is not surprising that they would be present in EVs, although it is unclear what orientation they would have with respect to the outer membrane of the EV and what impact they would have on transport. Can the organelle of origin of the BK channels be determined? Are the EVs positive for lysosomal markers?

We validated our EV preps for Mitochondrial marker (TOMM20), plasma membrane marker (E-cadherin), and lysosomal marker (LAMP-1) using western blots. The EVs prep showed no intracellular organellar contamination and were positive for EV markers (CD81 and HSP70). EVs were not positive for lysosome markers in our hands.

Fig.1: It is not clear that EV membranes would maintain polarization in extracellular solutions, particularly since BKCa channels, if present, should be highly activated in the high Ca^{2+} external environment. Moreover, it is debatable as to whether ion gradients could be maintained by ion pumps in the absence of ATP. Was the current transient recorded in the presence of EVs abrogated in the presence of a K^{+} ionophore or NaKATPase inhibitor? The evidence underpinning the validity of the NFE method is not provided

We value the reviewer's feedback on the K^{+} homeostasis within EV biology. We performed the ATP measurements from the plasma-derived EVs from humans. The data suggests the measurement of ATP were detected in the freshly isolated EVs (supplementary Fig. 4).

In Fig. 1F, the bilayer experiments show a channel that does not seem to be very K^{+} selective. With the K^{+} gradient mentioned (described as 500:50 mM in the legend, but 150:50 in the results section), the reversal potential should be in the neighborhood of -60 mV but it is closer to -20mV in panel 1F. Does this reversal point shift as expected when the K^{+} gradient is changed? Were these experiments repeated more than once?

All the bilayer studies were performed with asymmetrical K^{+} solutions containing 500:50::*cis:trans* KCH_3SO_3 with 50 μM CaCl_2 . We acknowledge that our E_r is less negative, but similar E_r in bilayers are reported earlier (DOI: 10.1380/ejssnt.2007.1, PMIDs: 21325511, 9547384, 10482751, 15968424). The reversal potential of the BK_{Ca} channel are dependent on the K^{+} concentration across the membrane [10482751], the presence of Ca^{2+} ions in the recording solution [9547384], and the regulatory β -subunit [30742788].

Fig. 3: Beat rate is not a direct measure of cellular injury, as any change to the resting membrane potential or pacemaker currents could alter this parameter. Indeed, activation of K^{+} channels would also affect this parameter. Thus, the changes shown in Fig. 3H-J

are of uncertain significance. Moreover, the assumption that altered miRNA profile has anything to do with the protection in EV-treated cells is not proven in this study. hiPSC-derived myocytes are usually quite tolerant of low levels of H₂O₂ (i.e. 10uM). Some additional measure of cell injury should be measured, for example, LDH release into the medium.

We have added the LDH release measurements in the different treatment groups in Fig.3 L of the revised manuscript.

Fig. 4. Fibrosis is measured from random sections post-MI. These measurements can be highly variable and should take into account multiple sections taken systematically. The lower panel of 4F shows echocardiograms from *Kcnma1*^{-/-} mice, but the images are very strange, with both walls moving in the same direction during the heartbeat. Is this cardiac dyssynchrony? How can one estimate fractional shortening reliably?

We have updated Fig. 4 to include a better representative echocardiogram trace. We performed an unbiased double-blinded data analysis on the Masson Trichrome stained section to evaluate the percentage of fibrosis between our experimental groups.

We thank all the reviewers for their highly useful comments and suggestions. We have added specific responses in blue. We have also modified the manuscript to reflect these changes.

Reviewer #1 (Remarks to the Author):

In the revised version the authors addressed most of my comments. Importantly, they performed the requested Western blots and surprisingly found no contamination of the EVs by mitochondrial membranes. However, in Suppl. Table 1. All 3 isoforms of VDAC as well as the mitochondrial calcium uniporter (listed among transporters, please change), known to be specifically located only in mitochondrial outer and inner membranes, respectively, are found. How can this be explained?

In Suppl. Fig 2 the authors forgot to add the number of experiments in the legend. Please correct. Although the authors answered most of my questions, my feeling is that they did not properly address some of the issues raised by the other reviewers, specifically reviewer 3. For example, the question of the channel orientation and the calcium-induced high activity remain open.

A. We would like to thank reviewer 1 for their valuable comments. The supplementary table is generated from an in silico analysis of all the data published in the literature and curated by Exocarta. The evidence ranges from proteomics to mRNA signature. Most of the channels and transporters listed are also located in other cellular organelles, including the plasma membrane, mitochondria, nucleus, lysosomes, and endoplasmic reticulum. There are two possible explanations for VDACS to be present in EVs:

1. EVs are known to possess mitochondria and mitochondrial membranes (10.1038/s41467-023-40680-5 and 10.3390/ijms24098181). In addition, mitochondrial membranes contribute to EVs (PMID: 34751216).
2. VDACS are known to be present in plasma membrane of cells. (PMIDs: 15861186 and 20184885.)

In both situations, VDACS or Ca^{2+} uniporter will end up in EVs.

B. We have also added n number in supplementary figure 2.

C. We agree that the orientation of BK channels is not yet determined in EVs. We contemplated both possibilities of BK orientation. I) C term containing Ca^{2+} sensing RCK domains facing outside EV which will immediately activate BK channel as Ca^{2+} concentration in extracellular environment is in mM and II) C term containing Ca^{2+} sensing RCK domains facing inside the EVs. In this case, Ca^{2+} gradient will push Ca^{2+} to EVs, which will also activate BK channels. In both scenarios, BK will remain highly active as stated by the reviewer.

Reviewer #2 (Remarks to the Author):

The authors have satisfyingly address all my comments.

We sincerely thank reviewer 2 for their valuable suggestions and comments.

Reviewer #3 (Remarks to the Author):

The authors have responded to the comments with some additional data to improve the manuscript.

Specifically: 1) They looked for markers of lysosomes and mitochondria and did not find any, indicating that the origin of the Bk channel was not likely to be from these organelles; 2) they determined that ATP was present in the EVs. This did not exactly answer the question about the origin of polarization of the EVs (the effects of a Na pump inhibitor or K ionophore might have been more informative from a functional point of view); 3) They did some hand waving about K selectivity, but did not vary the gradient to see if the reversal potential is shifted with the K gradient; 4) They added LDH measurements for the miRs, which is helpful; and 5) They found better echo recordings, which is an improvement.

In general, the reviewers have been responsive, although the precise role of the ion channels and transporters in EV ion homeostasis is somewhat unclear.

We sincerely thank the reviewer 3 for helpful comments and suggestions. Our manuscript has improved a lot with all the feedback. Here are the additional responses for specific questions raised by the reviewer 3.

- A. Since ATP is present in EVs, it raises the possibility of the existence of pumps and transporters in EVs, which will be tested in the future as the focus of the manuscript was to indicate functional ion channels do exist in EVs.
- B. BK channels are highly selective for K ions. Our recording solution contains K-methanosulfate, and the presence of currents indicates that they are K currents. Also, the negative reversal potential supports the conclusion for K ion conductance.
- C. We agree with the last point that the precise role of ion channels and transporters in EV ion homeostasis is somewhat unclear. Though this is the first manuscript to support the existence of functional ion channels, once more ion channels and transporters are discovered, we should be able to decipher their importance in regulating ionic homeostasis.